# Wireless Photometry Prototype for Tri-Color Excitation and Multi-Region Recording

**DOI:** 10.3390/mi13050727

**Published:** 2022-04-30

**Authors:** Aatreya Chakravarti, Amin Hazrati Marangalou, Ian Matthew Costanzo, Devdip Sen, Mirco Sciulli, Yusuke Tsuno, Ulkuhan Guler

**Affiliations:** 1Department of Electrical and Computer Engineering, Worcester Polytechnic Institute, Worcester, MA 01609, USA; achakravarti@wpi.edu (A.C.); ahazratimarangal@wpi.edu (A.H.M.); imcostanzo@wpi.edu (I.M.C.); dsen@wpi.edu (D.S.); msciulli@wpi.edu (M.S.); 2Department of Integrative Neurophysiology, Graduate School of Medical Sciences, Kanazawa University, Kanazawa 920-8640, Japan; tsuno@med.kanazawa-u.ac.jp

**Keywords:** optical excitation, optical recording, neural interface, wireless power transfer, wireless photometry, tricolor

## Abstract

Visualizing neuronal activation and neurotransmitter release by using fluorescent sensors is increasingly popular. The main drawback of contemporary multi-color or multi-region fiber photometry systems is the tethered structure that prevents the free movement of the animals. Although wireless photometry devices exist, a review of literature has shown that these devices can only optically stimulate or excite with a single wavelength simultaneously, and the lifetime of the battery is short. To tackle this limitation, we present a prototype for implementing a fully wireless photometry system with multi-color and multi-region functions. This paper introduces an integrated circuit (IC) prototype fabricated in TSMC 180 nm CMOS process technology. The prototype includes 3-channel optical excitation, 2-channel optical recording, wireless power transfer, and wireless data telemetry blocks. The recording front end has an average gain of 107 dB and consumes 620 μW of power. The light-emitting diode (LED) driver block provides a peak current of 20 mA for optical excitation. The rectifier, the core of the wireless power transmission, operates with 63% power conversion efficiency at 13.56 MHz and a maximum of 87% at 2 MHz. The system is validated in a laboratory bench test environment and compared with state-of-the-art technologies. The optical excitation and recording front end and the wireless power transfer circuit evaluated in this paper will form the basis for a future miniaturized final device with a shank that can be used in in vivo experiments.

## 1. Introduction

The fundamental technique of neuroscience is to record neuronal activity during specific tasks or behaviors in animals. One of the major indicators of neuronal activity is calcium signals. Because of the development of genetically encoded calcium indicators (GECI), we can record calcium signals of specific neuron cell types in vivo by using green calcium indicators [1] or red calcium indicators [2]. These indicators fluoresce at a specific wavelength of light when optically excited. The intensity of the emitted light is indicative of the concentration of calcium.

In addition to the calcium indicators listed above, in recent years specific sensors have been created to detect other neurotransmitter signals, such as glutamate [3], acetylcholine [4], or dopamine [5,6]. Although measuring a single neurotransmitter provides insight into its role in the brain, monitoring more than one neurotransmitter in different brain regions expands our understanding of the function of the brain.

Although the recently developed techniques of fiber photometry allow us to record with two or more colors [7] or from multiple brain regions [8,9], the primary constraint of the current fiber photometry is the tethered structure. The tethered system gives some degree of restraint and stress to animals, preventing us from performing longitudinal or multiple animal experiments during social interactions. To ensure animals’ total freedom during the experiment, we present a prototype wireless photometry system along with its evaluation board to validate the functionality of the miniaturized integrated circuits (IC).

The proposed prototype is capable of tri-color optical excitation and recording and is equipped with both wireless power transfer and data telemetry. To record neuronal calcium activity in two brain regions, we envisioned a single shank with two photodetectors (PDs) and three light-emitting diodes (LEDs) at specific locations. However, in this early prototype, we focus on validating the integrated circuit performance rather than a complete system with a shank. Hence, we employed the PDs and LEDs on the evaluation board for testing purposes.

In Section 2, we frame a target application that can benefit from optical excitation and recording from two different brain regions. Section 3 discusses the proposed system design and architectures for lower level circuit elements. Section 4 demonstrates the measurement environment and results. Following this, a comparison with state of the art and an outline of future work required to convert this prototype circuit into a final device capable of in vivo measurements is discussed in Section 5. Finally, Section 6 concludes the article.

## 2. Target Application: Cholinergic Monitoring

One possible application of the two-brain-region tri-color optical excitation and recording system is monitoring cholinergic neuron activity. This monitoring application is motivated by the Cholinergic Hypothesis, proposed by [10], which theorizes that the cholinergic neurotransmitter pathway is linked to memory loss symptoms like those seen in patients suffering from Alzheimer’s disease. The disruption of this pathway manifests as low concentrations of calcium ions (Ca${}^{2+}$) in the hippocampus and basal forebrain.

To measure both brain regions simultaneously without interference, the GCaMP [1] fluorescent sensors expressed via viral vectors, hereon referred to as viral sensors, can be used in conjunction with the RCaMP [2] sensor. A proposed shank insertion angle and shank properties are recommended in this paper to utilize both sensors. The relative LED and PD locations, as given in [11], were determined from the parasaggittal mouse brain dimensions provided in [12]. The shank insertion angle perpendicular relative to the surface of the animals’ brain should be at a ∼27° angle. The ACh excitation and recording components should be placed 3 mm down from the headstage to intersect the hippocampus region of the brain. The Ca${}^{2+}$ components should be placed farther down the shank 8 mm from the headstage to intersect the basal forebrain. Figure 1 is the envisioned use of our system with a shank. However, fabrication and utilization of the shank for in vivo experiments is planned for future research.

## 3. System Design

Figure 2 shows the block diagram of the overall system, illustrating the integration of various sub-blocks. The sub-blocks can be categorized based on the four main system functionalities: optical excitation (LED driver), optical recording (analog front-end), wireless power and data telemetry, and power management. In the following subsections, the topologies of each sub-block will be explored in detail.

### 3.1. LED Driver

The LED driver block comprises a decision-making circuit, a MOSFET power transistor (${PWR}_{MOS}$) integrated on the application-specific integrated circuit (ASIC), and a microcontroller unit (MCU) deployed on the printed circuit board (PCB), as shown in Figure 3. The MCU (ATTiny 84; Microchip Technology Inc., Chandler, AZ, USA) generates the pulse width modulated (PWM) input signal, which determines the LED blinking frequency and duty cycle, both parameters that can be modified by programming the MCU. The decision-making circuit consists of a NAND-NOR stage that either inverts or passes the PWM input dependent on the enable signal (ENBL) value. The gate driver consists of a chain of inverters that increases the PWM signal strength to drive the gate of a large MOSFET power transistor, ${PWR}_{MOS}$.

This configuration is inspired by a boost converter topology where the NMOS transistor acts as a switch, periodically charging and discharging the inductor. A low-resistance path to the ground is created when the transistor is on, which builds up the magnetic field around the inductor. When the transistor is off, the built-up field collapses, dumping current into the $V_{LED}$ node, boosting the voltage above the LED forward voltage, and causing current conduction.

As noted in Section 1, the device has three channels of optical excitation. However, the emission wavelengths of the ACh and ACh${}_{iso}$ signals are both 520 nm. Therefore, to allow recording of these signals, a power transistor is used at the output of one LED driver to time multiplex the excitation.

### 3.2. Analog Front-End (AFE)

The AFE stage includes a transimpedance amplifier (TIA) followed by a fully differential amplifier (FDA), depicted in Figure 4. The feedback network sets the FDA gain, where each feedback element consists of a pseudo resistor that allows for variable gain. The resistor value can be modulated with the $V_{R1C}$ and $V_{R2C}$ control signals. A photodiode model, consisting of a current source, a shunt resistor, and a shunt capacitor, is utilized at the input of the TIA for modeling and simulation purposes. The ideal current source represents a charge flow from cathode to anode when the light is incident on the photodiode. The shunt components take into account the parasitic resistance of the component leads and the junction capacitance.

The TIA used in this circuit is an exploratory design of a differential photodiode measurement configuration based on circuits presented in [13,14]. The topology contains a current sensing input stage that sets the bias level of the photodiode through local feedback. The bias voltages from positive and negative inputs ($V_{BIASP}$ and $V_{BIASM}$) are provided externally from an off-chip source to the positive input of operational amplifiers (OPAMP) A1 and A2, shown in Figure 5. Due to the OPAMP’s high open-loop gain, externally applied voltages at the positive input of A1 and A2 show up at the negative inputs through the “local” feedback loop connected to each of the photodiode terminal nodes. Therefore, the bias voltage across the photodiode is the difference between the positive and negative externally applied voltages, V${}_{BIASP}$ and V${}_{BIASM}$. To achieve a reverse bias, V${}_{BIASP}$ is set to a higher value than V${}_{BIASM}$.

A second, “main” feedback loop mirrors the current in the photodiode to be measured by the output stage transimpedance amplifier. When $I_{PD}$ increases, the drain current of M1 decreases, and thus the voltage at the gate of M3 decreases. The increase in gate-source voltage increases the current on M3, pulling the node N1 down. This action increases the current sourced by M5, balancing the currents at the input $V_{INP}$. The opposite occurs for the negative TIA path. When photocurrent increases, the voltage at the drain of M2 increases, decreasing the current on M4. This pushes the voltage at node N2 up, decreasing the current supplied by M6.

The output stages perform the final TIA function by converting the photocurrent into the voltage at outputs, V${}_{OUTP}$ and V${}_{OUTM}$. In the main feedback loop, when the voltage at node N1 is pulled down, M9 sources more current, increasing V${}_{OUTP}$ through mirroring the photocurrent trend. The opposite occurs at V${}_{OUTM}$, where increased photocurrent decreases the output voltage. Finally, the common-mode feedback (CMFB) circuit ensures that the output DC bias remains at V${}_{DD}$/2 by averaging the outputs and comparing the averaged output to the common-mode voltage (V${}_{CM}$). If the averaged voltage exceeds V${}_{CM}$, M7 and M8 are turned on to pull the voltages at V${}_{OUTM}$ and V${}_{OUTP}$ down via simple negative feedback. Subsequently, V${}_{OUTM}$ and V${}_{OUTP}$ will be fed into the analog-digital conversion (ADC) block in the MCU through AFE${}_{OUT}$ outputs to digitize the analog voltage. Digitized data can be stored or transmitted through the backscattered inductive link.

### 3.3. Power Management

The V${}_{REC}$ from the WPT stage is fed into a low-dropout regulator (LDO), outputting a stable 1.8 V supply. The LDO topology consists of a simple error amplifier configured in a negative feedback configuration. The resistive voltage divider ratio determines the regulated voltage output. Internal to the error amplifier is an operational transconductance amplifier (OTA) followed by an NMOS power transistor. An OTA is preferred to an OPAMP because there is no first-stage dominant pole, reducing the number of poles and improving stability. Finally, a large capacitor is placed at the load to dampen any peaking and store energy. The 1.8 V LDO output is fed into a beta multiplier (BMR) circuit followed by a bias generator to produce various reference levels required to bias various circuit in the system. The interconnection of blocks is depicted in Figure 6.

### 3.4. Wireless Power Transfer and Data Telemetry

A 30 × 13 mm (W × H) planar antenna is deployed on the prototype to enable both wireless power and data transmission. A rectifier circuit is designed to convert radio frequency (RF) signal to rectified DC voltage, supplying the LDO to provide a stable 1.8 V output DC voltage to the overall circuitry. The proposed active rectifier employs 3 V thick-oxide transistors fabricated in a 0.18 μm CMOS technology. Figure 7 illustrates the architecture of the designed full-wave active rectifier, adopted from [15], in which two high-speed comparators and one start-up circuit [16] contribute to improving the power conversion efficiency (PCE) of the rectifier.

The rectifier starts operating in the passive mode before V${}_{REC}$ reaches a certain voltage level. Once that level is reached, rectified V${}_{REC}$ voltage can properly operate the comparator. This operation leads to a transition in the rectifier from the passive mode to the active mode, in which a high-speed comparator controls the switching activities of the power PMOS transistors, P1 and P2. Moreover, a start-up circuit enables two complementary signals, SU and SU${}_{B}$, that help the transition between passive and active modes by enabling or disabling the comparator and forming the diode connection through an auxiliary transistor on the main power PMOS transistors. When V${}_{REC}$ is not high enough, SU goes high, and the comparator is disabled. Then, SU${}_{B}$ turns on transistors M5 and M6, which creates diode-connected P1 and P2 transistors, respectively; with that, V${}_{REC}$ starts charging up. When V${}_{REC}$ reaches the threshold at which the comparator starts operating, SU goes low, and SU${}_{B}$ goes high, resulting in transistors M5 and M6 turning off and enabling the comparator. Therefore, P1 and P2 transistors operate as active switches.

The high-speed comparators improve the PCE of the rectifier. The switches should turn on quickly to conduct the forward current thoroughly to the load. When V${}_{INP}$ is higher than V${}_{REC}$, the output of the first comparator (CMP1) goes low and turns P1 on. Therefore, the input current flows through P1 to charge up the V${}_{REC}$ node. When V${}_{REC}$ is larger than V${}_{INP}$, the switches should turn off fast to avoid back-current flowing from the load to the input. In other words, when V${}_{REC}$ is larger than V${}_{INP}$, current flows in the reverse direction through the drain of P1 to the source, causing degradation in PCE. As a result, the comparator output goes high to turn P1 off. To further decrease PCE degradation, M1 (M3) and M2 (M4) provide dynamic body biasing to prevent current flow into the bulk, causing a power loss. As the highest potential changes due to the nature of the RF sinusoidal signals, in the case of a fixed body-bias connection, the voltage at the source of the P1 and P2 may be larger than the voltage at the bulk, i.e., V${}_{INP}$ is greater than V${}_{REC}$, enabling P-N junction’s forward bias, causing current to be stolen from the main path.

The data telemetry functionality is implemented through backscatter communication using an on-off keying (OOK) modulation scheme. For circuit implementation, two series NMOS transistors are placed between the two terminals of the receiver antenna. When the transistors are turned on, they effectively short the antenna terminals. To transmit a “1” bit, the transistors are turned off. Due to the transmitter-receiver coupling, when the antenna is shorted, a temporary low voltage signal is observed at the transmitter. We can transmit a bit pattern by shorting and opening the antenna terminals through these transistors. In this system, the microcontroller digitizes the analog signals recorded by the analog front-end and feeds that data to an onboard gate driver. A gate driver pulls the gate of the power transistors to the ground once they are turned on and vice versa.

## 4. Results and Discussion

We tested each block on the prototype to verify the system’s operation. Figure 8a displays the PCB used to evaluate the separate blocks on the system. Figure 8b illustrates micrograph images of Chip 1 and Chip 2, highlighting the layout location of each sub-circuit on the evaluation board.

### 4.1. LED Driver Measurements

We utilized the microcontroller to generate 50% duty cycle, 1 kHz PWM signals, fed to the on-chip gate driver. We observed the V${}_{LED}$ node of the LED driver, the anode of the LED, shown in Figure 3, and the measurement results are displayed in Figure 9a. When the switch is off, the LED voltage remains at 1.8 V. When the switch is on, the inductor dumps the built charge at the LED anode and the voltage at V${}_{LED}$ jumps to 2.5 V. This peak voltage exceeds the 2.1 V forward voltage of the LED, thereby causing a 20 mA peak current to flow through the LED.

Thermal characterization of the utilized picoLEDs was carried out with an IR thermal camera (Flir One; Teledyne Inc., Thousand Oaks, CA). The temperature variation over a five-minute period with a 1 kHz, 20% duty cycle driving signal, generating a forward current of 20 mA in the 405 nm, 470 nm, and 560 nm LEDs, was 0.9, 0.9, and 1 ${}^{\circ }$C, respectively. The peak temperature variation across this range meets the 1 ${}^{\circ }$C limit outlined by [17]. Optical characterization of the picoLEDS was done with an optical power meter (PM101; Thorlabs Inc., Newton, NJ, USA). The range of optical power was 1 μW to 300 μW.

### 4.2. AFE Measurements

During in vivo usage, neuron light emissions from the brain of a live animal would be incident on the photodiode. In this paper, to replicate an in vivo light emission in a bench test, live animal model emissions were measured using a fiber photometry system (FP3001; Neurophotometrics Ltd., San Diego, CA, USA). The data were collected from a wild-type mouse (C57BL/6J) expressing a green calcium indicator (jGCaMP7s; [1]) in the dorsal hippocampus. More than two weeks before the recording, the virus vector (AAV-CAG-jGCaMP7s, 0.3 μL) was injected into the dorsal hippocampus (AP: −2.0, lateral: 1.3, depth: 1.1 mm from the surface) and an optical fiber (200 μmm core, N.A. 0.39, 1.25 mm ferrule) was implanted 0.1 mm above the virus injection site. Two LEDs (415 nm: 80 μW, 470 nm: 160 μW) were pulsed at 30 Hz in an interleaved manner. The emitted green fluorescence was captured by a CMOS camera sensor with a custom-written Bonsai code and processed offline by MATLAB. The data provided from the fiber photometry study described above were then inputted in a function generator and used to drive external through-hole LEDs such that the LED emission would mimic those produced by live animal neurons in a bench test setting. The light from these LEDs was then made incident on the photodiode, and the resulting AFE output was observed. The brain data used to drive the through-hole LEDs used in our bench test have peak optical power in the 1–10 Hz range. The resulting emissions were detectable with the proposed photodiode down to as low as 10 nA photocurrents. Figure 10a,b illustrates the testbench setup. The positive and negative inputs were excited independently in this experiment.

As shown in Figure 9c, the external LED emission waveform is replicated at both AFE outputs, albeit inverted for the negative output. The difference between the positive and negative output waveforms demonstrates the differential functionality. These waveforms provide verification of the performance of the AFE circuit. Additionally, Figure 9b displays the measured results of the AFE bandwidth and gain.

### 4.3. Power Management Measurements

To ensure correct reference voltage values were being generated, we tested both the BMR and bias generator along with the supply voltages generated by LDOs. This is of crucial importance because the biasing provided by these blocks is used throughout the system to ensure transistors are in the desired mode of operation. Figure 9d exhibits the voltages for all the outputs of the power management blocks.

### 4.4. WPT and Data Telemetry Measurements

Figure 11 illustrates the measurement testbench for the fabricated active rectifier and the wireless power transfer link. We utilized a balun to obtain the differential input signals, V${}_{INP}$ and V${}_{INN}$, without shorting $V_{INN}$ to the common ground through the BNC terminal of the function generator. Figure 12a depicts the transient waveforms of the inputs and output of the rectifier, operating at a frequency of 13.56 MHz, recorded by an oscilloscope.

We also designed a transmitter board (Tx) with a power amplifier and an antenna driven with a 13.56 MHz square wave to test the WPT circuit functionality. We observed the resulting antenna-coupled sine wave at the receiver antenna in Figure 12b. The original transmitted 5 V peak-to-peak sine wave was attenuated to a 3 V waveform at the receiver. This attenuation can be explained by the <1 coupling coefficient, which captures the loss due to transmission through the intervening medium. The resulting received sine wave was fed into the rectifier.

As described in Section 3.4, wireless data transmission was enabled through backscatter communication, taking advantage of the antenna coupling that provides wireless power to the system. To test the backscatter functionality, the onboard microcontroller was programmed to produce a 50% duty cycle square wave of various frequencies starting at 10 Hz going up to 1 MHz, which was fed to the gate driver. Figure 12b displays the modulation of the transmitter antenna signal, resulting from periodically shorting the receiver antenna using a power switch.

The rectifier characterization through PCE measurements was carried out using a 10 Ω current-sensing resistor in series with the input of the rectifier. The resulting waveform was measured using an oscilloscope, and the average of the product of the differential input voltage and sensed current was taken to calculate the average input power. In order to verify the achieved PCE results, a vector network analyzer (VNA) based technique explained in [18] was used. By measuring the scattering parameter (S${}_{11}$), the average input power of the rectifier can be calculated by subtracting the reflected power.

We characterized the rectifier circuit through the following experiments. First, we measured V${}_{REC}$ while sweeping R${}_{L}$ from 200 Ω to 10 kΩ, depicted in Figure 13a. The rectifier provides more than 3 V voltage at 1 kΩ and 3.4 V voltage after 4 kΩ. This measurement was performed with a V${}_{IN,Peak}$ of 4.2 V at 13.56 MHz. Second, we measured PCE when sweeping R${}_{L}$. We kept the same measurement conditions. Figure 13b illustrates that the highest PCE of 63% was achieved at R${}_{L}$ = 500 Ω. At this load resistance, the rectifier provides an output power of 13.83 mW. The PCE degrades notably before 400 Ω and after 1 kΩ. Next, we investigated the PCE of the rectifier for different input power swept from 12.8 mW to 41 mW, depicted in Figure 13c. Finally, we swept the frequency of the input signal from 0.5 MHz to 16 MHz. The highest PCE of 87% was achieved at the frequency of 2 MHz, shown in Figure 13d.

## 5. Benchmarking and Discussion

The main novelty of this work lies in the three wavelengths of optical excitation and two wavelengths of optical recording. We compared the specifications of the proposed photometry system prototype to other optical neural interfaces in Table 1. A device capable of dual-wavelength excitation was presented in [19]. However, only one wavelength of light can be produced for optical excitation at a given time. The optical power is in the 1–300 μW range, which meets the range 5–50 μW [8] for the target green and red calcium indicators application. This is in contrast to optogenetics stimulation devices that need an optical power range of 1–10 mW for excitation of channelrhodopsin [20]. On the recording side, the gain of the proposed device is larger than other state-of-the-art devices. The bandwidth and power are comparable. Finally, the WPT and data telemetry circuitry implemented in this system uses a two-coil inductive. For the rectifier, a respectable PCE of 63% is reported. For the data telemetry, OOK backscatter communication is used, which is similar to the other publications presented in the table.

The publications listed in this benchmarking comparison present devices ready for in vivo experimentation. To advance the prototype device described in this manuscript to an in vivo-ready stage, miniaturization will be further emphasized in the follow-up device. This miniaturization will be achieved by combining the optical excitation and recording circuits in Chip 1 with the WPT circuit in Chip 2. In addition, the test points used for chip characterization will be removed. Finally, a shank with the LEDs and PDs will be added, freeing up more space on the headstage board. With a smaller physical circuit, an optimized planar antenna will be designed to encompass the combined chip and peripheral circuitry.

## 6. Conclusions

The main objective of this work is to design a device capable of: (i) simultaneous optical excitation of two viral sensors GCaMP and RCaMP, (ii) optical recording of emissions from two brain regions, and (iii) wireless operation to allow for the natural behavior of animals subjects. The first objective was met by using two LED drivers to allow for the optical excitation of three different wavelengths of light. On the recording side, the device was designed with surface mount photodiodes capable of measuring fluorescence signal emissions.A TIA was then used to convert the photocurrent generated as a result of neuron light emissions into a detectable voltage. Finally, the last feature was satisfied by including both WPT and data telemetry functionality in the device with an integrated antenna. Results from bench testing demonstrate that the circuit blocks for each of these three stages operate as designed.

## Figures and Tables

**Figure 1 micromachines-13-00727-f001:**
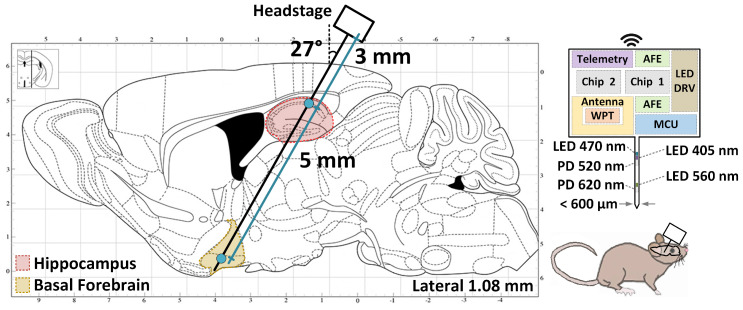
Parasaggittal mouse brain slice [12] with the highlighted hippocampus (red) and basal forebrain (yellow) demonstrating proposed shank positioning.

**Figure 2 micromachines-13-00727-f002:**
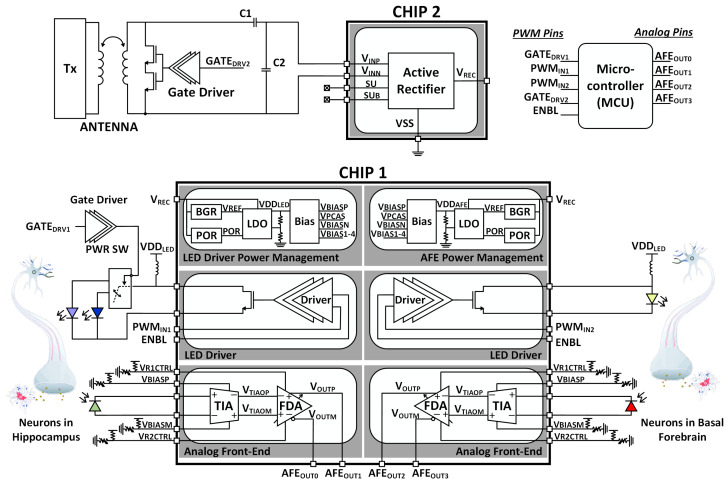
System block diagram.

**Figure 3 micromachines-13-00727-f003:**
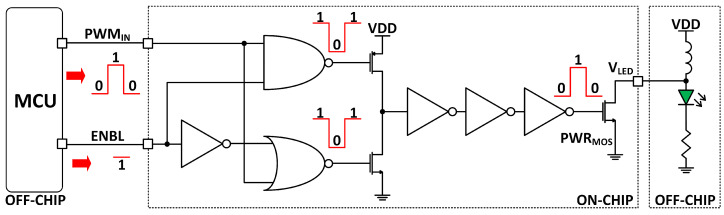
Schematic of the LED driver block.

**Figure 4 micromachines-13-00727-f004:**
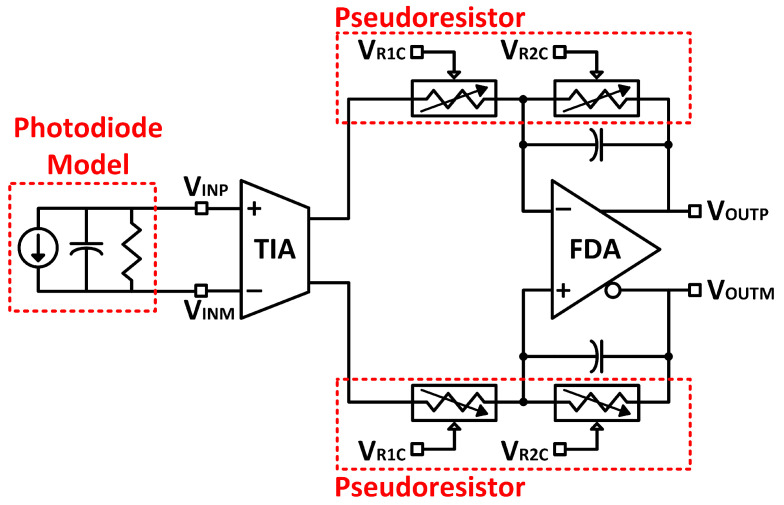
System-level schematic of the AFE block.

**Figure 5 micromachines-13-00727-f005:**
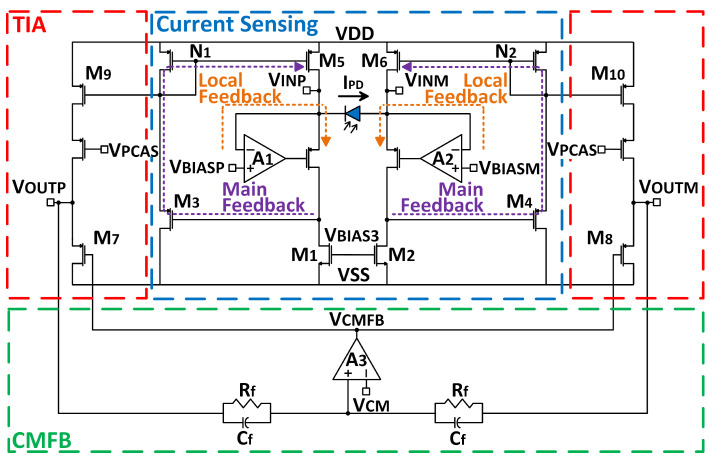
Transistor-level schematic of the TIA circuit.

**Figure 6 micromachines-13-00727-f006:**
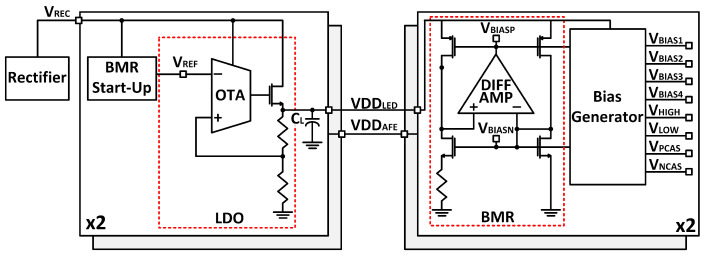
Transistor-level schematic of the LDO and BMR circuits.

**Figure 7 micromachines-13-00727-f007:**
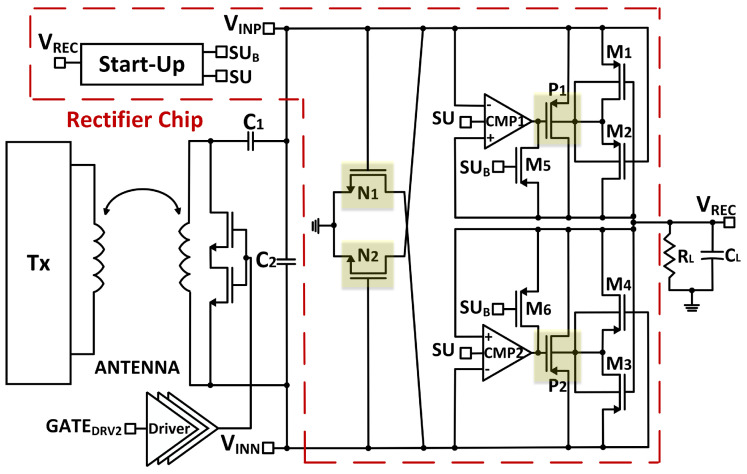
Transistor-level schematic of active rectifier.

**Figure 8 micromachines-13-00727-f008:**
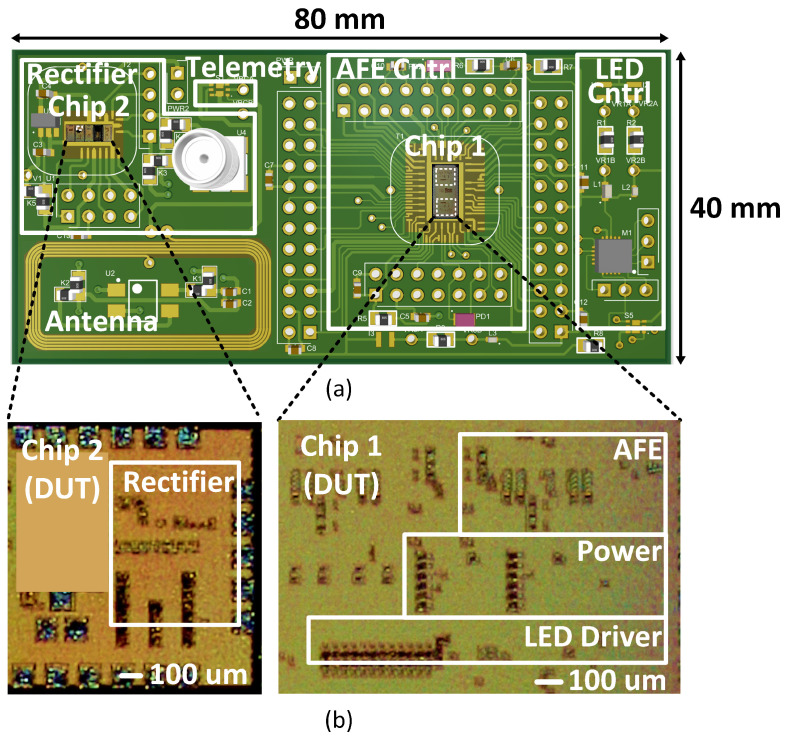
(**a**) Illustration of PCB with components, (**b**) micrographs of Chip 1 and Chip 2.

**Figure 9 micromachines-13-00727-f009:**
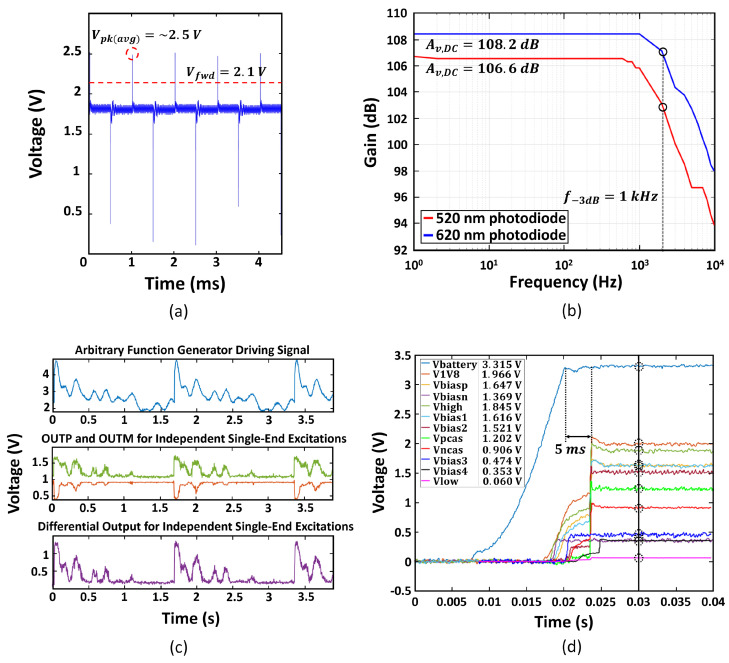
(**a**) Voltage at $V_{LED}$ node, (**b**) TIA bode plot for single-ended excitation, (**c**) waveform driving external LEDs, single-ended outputs, and differential output signal from AFE, (**d**) voltages generated by the BMR and the bias circuit.

**Figure 10 micromachines-13-00727-f010:**
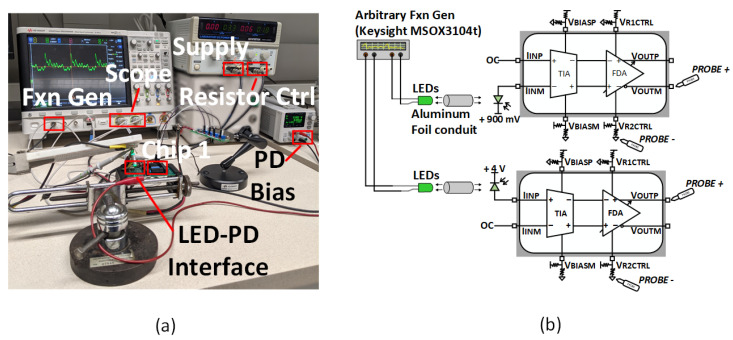
(**a**) Image of the testbench used for verification of AFE and LED driver functionality, (**b**) illustration of the testbench used for single-ended AFE excitation.

**Figure 11 micromachines-13-00727-f011:**
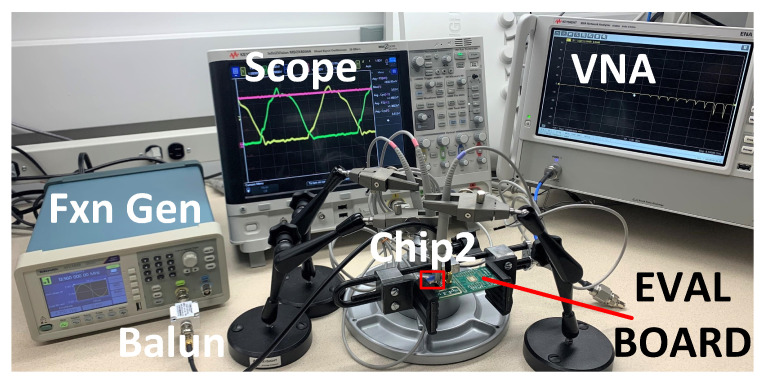
Testbench for measuring the active rectifier.

**Figure 12 micromachines-13-00727-f012:**
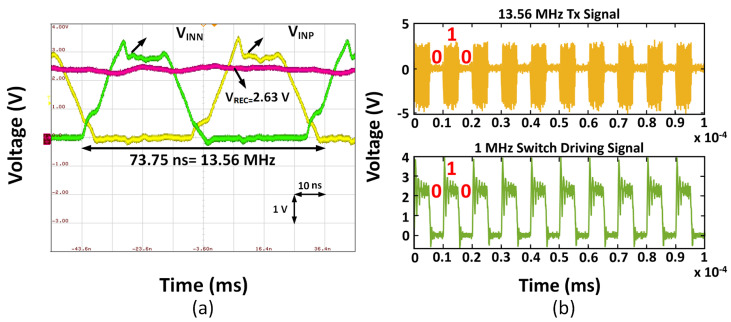
(**a**) Input and output waveform of the rectifier at 13.56 MHz, V${}_{IN,peak}$ = 4.2 V, R${}_{L}$ = 500 Ω, and C${}_{L}$ = 10 μF, (**b**) transmitter waveform modulation when power switch between receiver antenna inputs is driven with 10 kHz square wave.

**Figure 13 micromachines-13-00727-f013:**
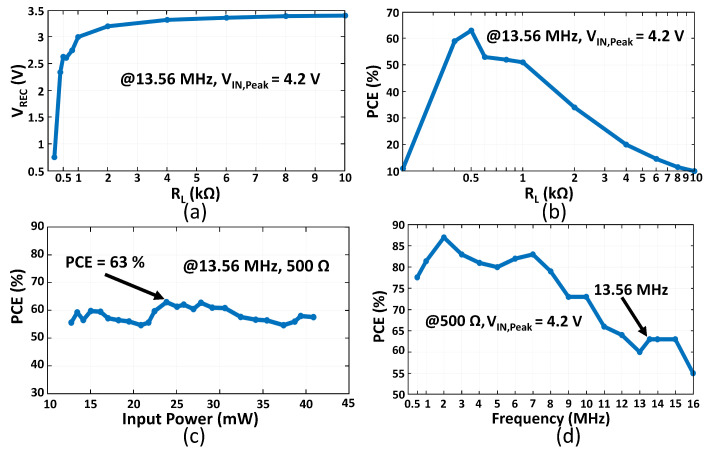
Rectifier measurement results (**a**) V${}_{REC}$ versus R${}_{L}$, (**b**) PCE versus $R_{L}$, (**c**) PCE versus input power, (**d**) PCE versus frequency.

**Table 1 micromachines-13-00727-t001:** Comparison with state-of-the-art systems.

Parameters	MDPI ’20 [21]	TBIOCAS ’20 [22]	TBIOCAS ’20 [23]	Nature ’18 [19]	MDPI ’22 [This Work]
Technology	0.35u	0.13u	0.35u	NM	0.18u
**Recording (AFE) **					
Modality	NA	Electrical	Electrical	Electrical	Optical
Gain (dB)	NA	48	55–70	NA	106–108
Bandwidth (Hz)	NA	1k	1–100, 10k	20k	1.1k
Power (uW)	NA	160	800	320	620
No. of channels	NA	2	16	32	2
**Excitation/Stimulation**					
Modality	Optical	Optical	Optical	Optical	Optical
Supply Voltage (V)	5	3.3	4	NM	1.8
Peak current (mA)	12	15	24.8	NM	20
Optical Power (uW)	NM	85.3	NM	NM	1–300
No. of colors	1	1	1	2	3
No. of channels	16	2	16	32	3
LED λ (nm)	470	488 568	460	405 635	405 470 560
**WPT and Telemetry**					
Structure	3-coil	2-coil	4-coil	NA	2-coil
Meas. PCE (%)	43	NM	82	NA	63
Load (Ω)/(μF)	NM/10	NM/200	NM/NM	NA	500/10
WPT Freq (MHz)	60	85.86	13.56	NA	13.56
Optimum $V_{REC}$ (V)	4.2	3.3	4.2	NA	2.63
Data Modulation	OOK	NM	OOK-PPM	NA	OOK

NA = not applicable, NM = not mentioned.

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
