# Peer review of "Wireless Photometry Prototype for Tri-Color Excitation and Multi-Region Recording"

_micromachines, 2022, doi:10.3390/mi13050727_

Round 1

Reviewer 1 Report

In this article, the authors described the design and validation of a wireless neural interface that perform 2-color photometry and optogenetic stimulation. While I’m not qualified to judge the circuit design of the device, I can provide opinions from a neuroscientist as a potential user of such device. Overall, the device has good merits if properly validated, but the results presented are hardly convincing. Several improvements should be made to make this article close to publishable.

  1. The device is in principle a wireless 2-color photometry with the capability for optogenetic stimulation, the mention of cholinergic neuronal activity in the title seems far fetched without further demonstration. So is the cholinergic hypothesis in the introduction.
  2. Figure 1 seems unrelated to this study. (Age and AD status are not important factors for the device presented. Not to mention the data presented in figure 1 is not part of the study)
  3. Figure 2 reproduced from ref5, which originally is partly reproduced from http://labs.gaidi.ca/mouse-brain-atlas/, yet proper citation is missing.
  4. The size of the device with a 30x13 antenna is bigger than state of art wireless neural interfaces. It is questionable if it can be used in free behaving mouse, as a mouse atlas is used in Fig.2.
  5. Line 251: What is the output of the LED and how is the power compared to typical range used in optogenetics/photometry?
  6. Line 258: What system are the fiber photometry measurements from? What are the typical power output range of LED used in this experiment? In general, the use of LED to mimic fluorescence photometry signal is not an accepted model.
  7. Line 284: To assess wireless power transmission, power transfer efficiency should be given as a function of distance.
  8. Validation in any biologically significant system is missing. Characteristics important for potential application such as SNR for optical recording need to be described. At minimal testing in an in vitro model system is required, and animal models are preferred

Reviewer 2 Report

Comment 1: “The statement about population growth in the first paragraph of the introduction (line 20-25) deviates from the core content of the article. Please suggest authors delete or simplify the relevant content to make the causal relationship between the context of the article clearer.”
Comment 2: “On page 2, authors introduced the current situation and experiments of demonstrating a link between the degradation of the cholinergic pathway and memory loss symptoms from line 40 to 68, which is unnecessarily lengthy. Please simplify relevant content and mainly describes the disadvantage of the previous work compared with the work in this paper.”
Comment 3: “It is suggested that the author should put the related contents of system integration and device implantation shown in Figure 2 into Chapter 2.”
Comment 4: “The system block diagram shown in Figure 3 is too complex for readers to understand. It is suggested that the author add the introduction of system workflow or simplify the block diagram, as shown on the right side of Figure 2.”
Comment 5: “Figure 4, Figure 5, Figure 6, Figure 7 and Figure 8 show the schematic diagram of each part of the circuit is the same as Figure 3. It is suggested that the author put these pictures in the supplementary materials.”
Comment 6: “The authors do not mention the parameters, manufacturing process, or physical images of the shank that can be implanted in the brain. This part of the content should be supplemented in detail in the form of text or pictures in Chapter 2.”
Comment 7: “They are 3 LEDs and 2 PDs on the shank as shown in figure 2. The propagation model and optical simulation results of three LED light sources in brain tissue should be provided by the author. Explain how the relative positions of LED and PD are determined.”
Comment 8: “In chapter 3, the author should add the optical and thermal characterization of LED and PD used in the experiment.”
Comment 9: “The content of the article is inconsistent with the innovation point. The innovation of this paper lies in the simultaneous measurement of ACh in the hippocampus and Ca2+ in the forebrain, please suggest authors provide the corresponding long-term and repeatable experimental results of ACh and Ca2+ in vivo or in vitro, otherwise, it is not convinced.”
Comment 10: “Chapter 5 mentions that the system meets the requirements of wireless devices and achieves the goal of allowing the natural behavior of animal subjects. However, the size of PCB as shown in figure 9 (a) is 80mm * 40mm, which is a very big size. How to realize the wireless operation of this device, please provide a relevant animal experimental model and surgical explanation.”
Comment 11: How about the stability of the wireless power transfer, and what is the maximum working distance? 
Comment 12: In Abstract: What dose " The integrated circuits are designed in TSMC 180nm process node" mean?  And this should be explain in the manuscript.

Other comments: 
Line 22, the uW should be μW
It's very inappropriate to quote the reference at the beginning of the sentence. 

Round 2

Reviewer 1 Report

The authors addressed some questions in the report. The lack of biologically significant system is still a major drawback. It's up to the editor's discretion if such very early prototype device should be published. 

Reviewer 2 Report

The authors replied to my question, and I suggest that the article can be accepted for publication.